# Implications of Fatty Acids for Age-Related Macular Degeneration: Evidence and Recommendations

**DOI:** 10.3390/cells14120937

**Published:** 2025-06-19

**Authors:** Shivantika Bisen, Nikhlesh K. Singh

**Affiliations:** 1Integrative Biosciences Center, Wayne State University, Detroit, MI 48202, USA; sbisen@wayne.edu; 2Department of Ophthalmology, Visual and Anatomical Sciences, School of Medicine, Wayne State University, Detroit, MI 48202, USA

**Keywords:** retina, fatty acids, AMD, retinal degeneration, omega-3 fatty acids

## Abstract

Age-related macular degeneration (AMD) is an ocular pathology in humans characterized by the buildup of lipid-rich extracellular deposits, which leads to retinal degeneration. In recent years, considerable effort has been made to observe the effect of dietary fatty acids on oxidative stress and inflammation. In continuation of this, much effort has been made to study the effect of dietary fatty acids on the pathogenesis of AMD. Although studies have shown that dietary fatty acids are effective against few forms of AMD, particularly wet AMD or neovascular AMD, no dietary lipids have shown any conclusive results for dry AMD or geographic AMD. It is therefore important to look for new lipids and lipoproteins that can be helpful in treating various stages of AMD. This article reviews the impact of saturated fatty acids (SFAs), monounsaturated fatty acids (MUFAs), and polyunsaturated fatty acids (PUFAs) on retinal health and the progression of AMD. Furthermore, this manuscript discusses all studies investigating the implications of fatty acids on AMD, which may be beneficial for future treatment strategies and dietary guidelines related to it. In conclusion, studies suggest that omega-3 PUFAs, particularly docosahexaenoic acid (DHA) and eicosapentaenoic acid (EPA), might provide protection against AMD, highlighting the necessity for additional clinical trials to evaluate their efficacy in the prevention and treatment of AMD.

## 1. Introduction

One of the most important clinical characteristics for predicting mortality, physical and mental performance, disease risk, and several other important health issues is age. Aging at the molecular level means that molecular processes become less efficient and accurate over time. Cells become less efficient due to these changes in metabolism, gene expression, and epigenetics. Numerous ocular disorders are associated with aging; the most prevalent one is AMD [1]. AMD impacts almost 19.83 million people in the United States and is recognized as the primary cause of blindness in developed countries [1]. Age-related macular degeneration is a multifaceted human ailment distinguished by the accumulation of lipid-rich extracellular deposits known as drusen, which further leads to localized inflammation and ultimately leads to retinal degeneration. AMD results from a complex interaction of genetic and environmental factors [2]. The early and intermediate phases of AMD are identified by the presence of drusen (extracellular deposits) and pigmentary abnormalities in the retinal pigmental epithelium (RPE). Another manifestation of the early and intermediate stages of AMD is lipofuscin, which is abnormal waste buildup in retinal pigment epithelium (RPE) cells, and reticular pseudo drusen (RPD), which are extracellular deposits between the photoreceptors and the RPE [3,4]. We typically categorize late AMD into two types: dry AMD, also called geographic atrophy (GA), and wet AMD, also known as neovascular AMD [3,4]. Figure 1 illustrates the graphical representations of the various stages of AMD. Approximately 80% to 90% of cases of AMD, encompassing intermediate and late-stage GA, are of the dry, atrophic type. This type of AMD shows a slow progression over time, with atrophic areas growing and merging [5]. Wet AMD is characterized by proliferation of aberrant choroidal veins beneath the RPE, particularly in the macula [5]. In wet AMD, vision loss happens faster because of abnormal arteries that leak and bleed into the macula. This causes abrupt vision impairment caused by macular edema and direct injury to the photoreceptors, which leads to scarring over time [6,7,8].

Fatty acids (FAs), a principal dietary component, have garnered considerable interest owing to their numerous effects on human health and diseases [9], with the retina being one of the most lipid-rich tissues [10]. Dietary fatty acids (FAs) include saturated fatty acids (SFAs), monounsaturated fatty acids (MUFAs), and polyunsaturated fatty acids (PUFAs). These fatty acids are essential energy sources implicated in lipogenesis, glycolysis, and protein synthesis [11]. In retinal phospholipids, PUFAs comprise 45% of the total phospholipids, SFAs constitute 37%, and MUFAs represent 10% [12]. Polyunsaturated fatty acids (PUFAs) are predominantly sourced from dietary intake and are advantageous for anti-inflammatory and antithrombotic mechanisms, in addition to supporting eyesight, cognitive function, and the metabolism of glucose and lipids [11]. Numerous research studies indicate that PUFAs, particularly omega-3 PUFAs, reduce the risk of AMD, highlighting a notable disparity between high and low consumption of docosahexaenoic acid (DHA) in preventing the onset of AMD [13,14,15,16,17,18], while a limited number of studies suggest no significant difference [19,20,21,22,23]. A study demonstrated that varying levels of omega-6 PUFA intake, both high and low, significantly influenced the reduction in AMD incidence [24]. Another study identified a notable disparity in the prevalence of AMD between groups with high-omega-6 and low-omega-6 PUFA intake [25], while three research studies concluded that varying levels of omega-6 PUFA intake did not affect the incidence of AMD [15,26,27]. Furthermore, research on the correlations between specific MUFA or SFA intake and the risk of AMD is limited, and the findings have been conflicting [20,24,26]. This review aims to provide a comprehensive understanding of how SFAs, MUFAs, and PUFAs influence retinal health and disease progression. Additionally, it will explore potential therapeutic implications and dietary recommendations that could mitigate the risk of AMD through targeted fatty acid intake.

## 2. Human Fatty Acids

Fatty acids are vital for the human body, as they perform various functions such as energy supply, cell membrane formation, and the improvement of cardiovascular health [27,28]. Fatty acids are categorized based on their chain length and the existence of double bonds. Fatty acids consist of a hydrocarbon chain of varying length, terminating in a carboxyl group. SFAs lack double bonds, MUFAs have only one double bond, while PUFAs possess two or more than two double bonds, as illustrated in Figure 2. The position of the first double bond relative to the terminal methyl group classifies polyunsaturated fatty acids (PUFAs) into omega-3 (double bond on the third carbon) and omega-6 (double bond on the sixth carbon) [27,28]. Omega-9 fatty acids are MUFAs characterized by the presence of the first double bond on the ninth carbon atom [27,28].

Essential fatty acids are LC-PUFAs, like linoleic acid (LA, omega-6 PUFA, 18:2) and α-linolenic acid (ALA, omega-3 PUFA, 18:3), which the human body cannot synthesize because humans lack specific enzymes (Δ15 and Δ12 desaturases) needed to produce them from oleic acid (omega-9 MUFAs, 18:1) [29], so they must be supplemented by diet. ALA is the precursor of all omega-3 LC-PUFAs, and LA is the precursor of all omega-6 LC-PUFAs [30,31]. LA is present in sunflower seeds, Brazil nuts, eggs, and some vegetable oils, including soybean, sunflower, or palm oil [30]. Red meat also contains omega-6 fatty acids. ALA is present in quinoa, chia seeds, and flaxseed oils, among other vegetable oils [31,32]. The algae that fish eat are the primary plant source of omega-3 LC-PUFA. Therefore, fatty fish are a beneficial source of omega-3 fatty acids for humans [28]. As people age, the levels of LC-PUFAs and VLC-PUFAs in the retina, retinal pigment epithelium (RPE), and choroid diminish, particularly in patients afflicted with conditions such as AMD and diabetic retinopathy [33,34]. Due to the accumulation of saturated fatty acids in retinal tissue with age, it is critical to maintain a healthy ratio of unsaturated fatty acids to SFAs for ocular health [35].

## 3. Saturated Fatty Acids (SFA) and AMD

Studies on the correlation between saturated fatty acid consumption and the risk of AMD are scarce, with findings being conflicting [28,36,37,38]. Research indicates that elevated saturated fatty acid consumption correlates with a heightened risk of AMD [33,39,40], while a limited number of research studies propose an inverse relationship [38,41]. Since β-oxidation of fatty acids provides energy in the retina [42,43], adequate fatty acid intake is crucial for maintaining retinal homeostasis and AMD prevention. However, studies have proposed a combined impact of fatty acid intake and patient genotype on AMD [44,45,46,47].

Research on the correlation between saturated fatty acid consumption and the risk of AMD has been sparse, with findings exhibiting inconsistency. Many studies have shown that eating more saturated fatty acids is linked to a higher risk of developing early-stage AMD [36] or any stage of AMD [22], but other studies found no clear connection to either early or late AMD [17,38,48,49,50]. The Tsuruoka Metabolomics Cohort Study found that eating more SFAs is linked to a lower risk of early AMD, using a definition of AMD from the Age-Related Eye Disease Study (AREDS) classification [26]. Yamagishi et al. [51] found that eating fewer saturated fatty acids is linked to a higher risk of stroke, especially certain types of brain bleeding, in Japanese people. The higher rates of intraparenchymal hemorrhage and lacunar infarction in people who eat less SFA might be caused by arteriolosclerosis [52], which is connected to much lower total blood cholesterol levels, mainly because of low SFA consumption [53]. At the same time, various studies have shown that abnormalities in the diameter and density of choriocapillaris may contribute to the pathogenesis of AMD [54]. Damage and problems in the choriocapillaris lead to a lack of oxygen in the outer retina, which may trigger the onset of AMD [55].

A recent study on the Japanese population has shown a significant inverse association of SFA intake with prevalence of AMD [20]. Compared to Western cultures, Asians are known to consume fewer foods containing saturated fatty acids [56]. According to the 2017 to 2018 National Health and Nutrition Examination Survey, American women aged 20 years and older consumed 24.7 g/d of SFAs, whereas American men consumed 33.0 g/d of SFAs [54]. Males aged 20 years and older consumed an average of 19.0 g/d of saturated fatty acids, while females consumed 16.4 g/d, according to the 2018 National Health and Nutrition Survey in Japan [57]. In the investigation of the Japanese population, the average SFA consumption was 17.6 g/d for men and 20.2 g/d for women. The amount of SFAs consumed by Japanese participants, especially men, was much lower than that of people in Western countries, which might help explain the different results found in the Japanese study [20] compared to studies of Western populations [22,36]. The studies indicate that the relationships between fatty acid consumption and AMD may vary between groups with distinct genetic origins or dietary habits. Consequently, the ideal dietary levels of fatty acids must be evaluated within the target population, taking into account their sources before concluding based on the results of the dietary population studies.

## 4. Monounsaturated Fatty Acids (MUFAs) and AMD

MUFAs are prevalent in nuts, avocados, and olive oil. Diets abundant in MUFAs demonstrate beneficial anti-inflammatory and cardiovascular effects while enhancing lipid profiles [58]. MUFAs demonstrate anti-inflammatory properties through multiple biological mechanisms, including peroxisome proliferator–activated receptor (PPAR)-mediated suppression of nuclear factor-kappa B (NF-κB) [59]. Nevertheless, research investigations studying the correlation between MUFA consumption and AMD have yielded inconsistent results. Several studies have indicated a beneficial impact of MUFAs on AMD [23,25], whereas others reported contrary findings [26,48]. A study demonstrated a correlation between MUFA and AMD in Portuguese participants, but not in American subjects [24]. In the United States, meat and dairy products are the primary source of MUFAs [60], whereas the Portuguese diet consists of substantial amounts of olive oil (high in MUFAs) and fruit, grains, seeds, and nuts (rich in MUFAs and omega-6 PUFAs) [61,62]. Studies have observed an inverse correlation between AMD and increased consumption of olive oil [63], Mediterranean diet [64], and nut intake [65]. Therefore, it may be concluded that the source of MUFAs may be relevant in explaining the disparity between the outcomes of various research studies. In Japan, the sources of monounsaturated fatty acids (MUFAs) are like those of saturated fatty acids (SFAs), and a Japanese study found that eating more SFAs and MUFAs is linked to a lower risk of age-related macular degeneration (AMD), which is more common in men [20]. These studies indicate that the source of MUFAs can play a crucial role in eye health. More studies are needed to understand the mechanisms through which MUFAs could help prevent AMD, especially in people who are at higher risk.

## 5. Polyunsaturated Fatty Acids (PUFAs) and AMD

### 5.1. Long-Chain (LC)-PUFAs and AMD

PUFAs are essential for various physiological processes in the human body, including cellular architecture and the regulation of inflammation. The human body primarily produces LC-PUFAs in the liver [66]. Despite the local synthesis of these fatty acids in the retina, it does not meet the retina’s need for LC-PUFAs [67]. The retina predominantly consists of five major fatty acids: palmitic acid (16:0), stearic acid (18:0), oleic acid (an omega-9 MUFA, 18:1), arachidonic acid (an omega-6 PUFA, 20:4), and docosahexaenoic acid (an omega-3 PUFA, 22:6) [34]. Among these, docosahexaenoic acid (DHA) is the most abundant fatty acid. Additionally, eicosapentaenoic acid (EPA, an omega-3 PUFA, 20:5) contributes to fatty acid metabolism and supports retinal function. Desaturases are enzymes that introduce double bonds into fatty acids, and delta-6 desaturase (D6D), expressed by the FADS2 gene, is one of the rate-limiting enzymes that convert a-linolenic acid (ALA, 18:3 n-3) to EPA/DHA and linoleic acid (LA, 18:2 n-6) to arachidonic acid (AA, 20:4 n-6). Studies have shown that changes in D6D activity modify fatty acid profiles, which are associated with inflammatory and metabolic diseases. Although research on the involvement of D6D in AMD is limited, it has been proposed that D6D is involved in fatty acid metabolism and lipid homeostasis [68]. Therefore, it can be suggested that D6D might have a role in AMD pathogenesis [69]. In summary, LC-PUFA metabolites can influence inflammation by producing either pro-inflammatory (omega-6 PUFAs) or anti-inflammatory (omega-3 PUFAs) fatty acids.

#### 5.1.1. Omega-3 PUFAs and AMD

AMD is characterized by the deterioration of the macula and its lipid-rich photoreceptors. However, PUFAs, mainly DHA, are less abundant in the macula compared to the peripheral regions. Numerous studies have indicated a beneficial impact of fatty acid consumption, particularly EPA and DHA, in the prevention and management of AMD [67].

Various studies have elucidated the beneficial effects of omega-3 PUFAs in AMD and identified four primary mechanisms by which omega-3 PUFAs modulate AMD [70]. Their initial concept postulated that omega-3 PUFAs replace omega-6 PUFAs in phospholipid membranes. PUFAs help cells resist oxidative stress by activating a protein called nuclear factor (erythroid-derived 2)-like 2 (NFE2L2). It has also been shown that via blocking matrix metalloproteinase production and activating the adiponectin pathway, PUFAs reduce choroidal neovascularization [70]. In addition, numerous studies revealed that omega-3-enriched fatty acids diminish ROS generation and RPE degradation in comparison to a diet enriched with saturated fatty acids. Consequently, it can be inferred that a deficiency of omega-3 PUFAs may worsen AMD by compromising these systems [70].

Several studies and reviews looked at how diets high in PUFAs affect different stages of AMD, including early, intermediate, and late phases. A review of nine studies found that eating omega-3 PUFAs is linked to a 38% lower probability of late AMD [50]. The Age-Related Eye Disease Study 2 (AREDS2) was the only one that raised questions about the protective benefits of omega-3 PUFAs [71], while the AREDS indicated that eating omega-3 PUFAs is linked to a lower risk of AMD. The disparity between the AREDS and AREDS2 studies may be attributed to the varying patient recruitment guidelines among these research studies, the dosages of EPA and DHA provided, and the thresholds for detecting effects [14,71]. A recent study has shown that consuming less than 1 mg/day of DHA or EPA has no effect on age-related macular degeneration (AMD), while a protective effect on AMD is observed at a consumption level of more than 1 mg/day [72].

Studies have shown that EPA serves as a precursor of various anti-inflammatory molecules such as leukotrienes, thromboxane, and prostaglandins [73,74]. DHA is also known to produce anti-inflammatory lipid mediators such as resolvins (D1 and D4) and protectins [75]. Additionally, DHA and EPA are also shown to reduce or inhibit the increase in IL-6, InterCellular Adhesion Molecule (ICAM-1), IL-1β, and VEGF, which are important for neovascularization in wet AMD [76,77]. Overall, EPA seems to be more proficient in modulating the balance of anti-inflammatory cytokines, whereas DHA appears to preferentially suppress pro-inflammatory mediators [78]. Recent findings indicate that animals administered deuterated DHA (DHA with hydrogen atoms substituted by deuterium) exhibited less oxidative damage and provided protection against geographic atrophy in a mouse model of iron overload. Recent findings indicate that animals administered deuterated DHA (DHA with hydrogen atoms substituted by deuterium) exhibited less oxidative damage and provided protection against geographic atrophy in a mouse model of iron overload. [79,80]. The studies present a compelling biological rationale for the use of DHA and EPA in the prevention and management of AMD, highlighting their anti-inflammatory, antioxidant, structural, and anti-angiogenic properties.

#### 5.1.2. Omega-6 PUFAs and AMD

Diets rich in omega-6 LC-PUFA have been associated with elevated inflammatory responses [81]. In a study comparing different groups, higher levels of omega-6 PUFAs were linked to more cases of exudative AMD, while higher levels of omega-3 PUFAs were linked to fewer cases [82]. This study suggests that balancing the intake of these fatty acids may be crucial for reducing the risk of conditions like AMD. Therefore, incorporating more omega-3-rich foods, such as fatty fish and flaxseeds, could be beneficial for overall eye health [81]. Omega-6 PUFAs, prevalent in Western diets, may elevate inflammatory markers linked to AMD. On the other hand, increased consumption of omega-3 PUFAs has been associated with a 12% reduction in the chance of developing AMD in individuals considered to be at high risk [83]. Diet plays a significant role in determining the amounts and balance of omega-6 PUFAs and omega-3 PUFAs in the bloodstream, which subsequently affects their levels and balance within the retina [84]. Additionally, with advancing age, the concentrations of PUFAs in the retina and RPE/choroid exhibit considerable variations. The changes are most noticeable in the very long-chain PUFA lipid family in eyes affected by AMD, showing a significant decrease in the retina, being undetectable in the RPE/choroid, and having higher ratios of omega-6 PUFAs to omega-3 PUFAs [34]. One reason omega-3 PUFAs are beneficial is that when EPA and DHA are added to cell membranes, they replace similar omega-6 PUFAs, mainly linoleic acid and arachidonic acid [85].

### 5.2. Very Long-Chain (VLC) PUFAs and AMD

VLC-PUFAs were first isolated and examined in bovine retinas by Aveldaño in 1987 [86]. VLC-PUFAs are presently categorized as fatty acids having the longest continuous carbon chain, exceeding 24 carbons. Their hybrid structure has a carboxylic end close to the main chain, which is common in saturated fatty acids, and an end further away that has cis double bonds separated by methylene groups [87]. ELOVL4 and other biosynthetic enzymes produce the majority of retinal VLC-PUFAs in situ and de novo through tandem elongation and desaturation processes. Retinal VLC-PUFAs are generated from the PUFAs with 3 and 6 double bonds [88,89], and scientists have found VLC-PUFAs containing as many as 13 double bonds in the tissues of vertebrates [90]. Vertebrate diets do not contain VLC-PUFAs and must consequently be synthesized locally, as the liver lacks the requisite enzymes [87]. VLC-PUFAs are limited and located solely in specialized tissues such as the brain, retina, meibomian gland, and reproductive tissues of vertebrates [91]. Besides animal tissue, VLC-PUFAs can be obtained by extracting and purifying them from cells that produce a lot of ELOVL4 or made in a lab using organozinc coupling methods that involve LC-PUFAs and a SAFs [92]. VLC-PUFAs are synthesized in the retina from EPA and arachidonic acid, with the help of the enzyme ELOVL4 [93]. The neural synaptic vesicles in photoreceptors contain VLC-PUFAs, which are important for keeping the membranes flexible, curved, and intact [94,95]. Genetic mutations in ELOVL4 cause Stargardt-3 disease (STGD3) [96,97,98]. Autosomal recessive mutations in both copies of the ELOVL4 gene cause serious problems that often result in early death because of reduced skin protection and brain function due to lower amounts of SFAs and VLC-PUFAs [84,99,100].

Although changes in the ELOVL4 gene are not linked to the risk of AMD, studies indicate that eyes from people with AMD [33] and those with diabetes [101] have lower levels of VLC-PUFAs and unusual omega-3/omega-6 ratios in both the peripheral retina and macula, probably due to problems with ELOVL4 function [102] and a lack of dietary precursors. Researchers have also linked a higher risk of AMD to the age-related decline in ELOVL2 function, which may impact VLC-PUFA levels [103]. A dietary study has found that Utah family members with STGD3 who eat a lot of fish (rich in EPA and DHA) show less severe symptoms than those who rarely eat fish [104]. A recent study suggests that AMD patients exhibit a photoreceptor-specific (rod) reduction in the omega-3 VLC-PUFAs [105]. The MALDI imaging of retinal sections has demonstrated a significant reduction in omega-3 VLC-PUFAs in AMD retinas (Figure 3). Numerous epidemiological studies have shown a lower incidence of advanced AMD among individuals who eat substantial amounts of fish in their diets [38,106,107]. Research regarding the impact of pure fish oil supplements on STGD3 and AMD has not demonstrated a reduction in disease severity, either due to the lack of appropriate VLC-PUFA precursors or the patients’ failure to adhere to the prescribed diet [106,108]. It can be suggested that if patients with STGD3 or AMD take enough VLC-PUFAs that the retina can use directly, it might help avoid the processes that ELOVL4 [107] and ELOVL2 [103] support, possibly slowing down or stopping degeneration. Studies have shown that administration of omega-3 VLC-PUFAs (C32:6) through oral gavage in mice shows its bioavailability in ocular regions and the retina, enhancing visual function [109,110]. Studies suggest that bioactive lipids produced from VLC-PUFA protect photoreceptors from oxidative stress [111,112]. Studies have shown that oxygenated derivatives of VLC-PUFA called “elovanoids” function as a survival signal during periods of oxidative stress [112]. The authors demonstrated that elovanoids (ELV-N32 and ELV-N34), dehydroxylated derivatives of omega-3 VLC-PUFAs (32:6 and 34:6), safeguard RPE cells from hydrogen peroxide-induced apoptosis by downregulating pro-apoptotic proteins Bax, Bim, and Bid and upregulating anti-apoptotic proteins such as Bcl-2 and Bcl-xL [112].

Patients with AMD had decreased VLC-PUFA levels and a smaller omega-3 to omega-6 PUFA ratio compared to healthy individuals. The limited availability and challenges in synthesizing VLC-PUFAs have hindered their investigation until recently. Furthermore, their restriction to specific tissues makes them difficult to obtain through a Western diet. As a result, studying the impact of these lipids on ocular health has proven challenging, particularly in people with vision problems like AMD and diabetic retinopathy.

Researchers are now exploring alternative methods to enhance the intake of VLC-PUFAs, including dietary supplements and targeted therapies. By understanding the role of these fatty acids in eye health, they hope to develop new strategies for preventing or managing conditions such as AMD and improving overall vision.

## 6. Future Challenges

AMD is a multifaceted disorder, and the precise starting causes are still a matter of contention. Various factors are responsible for causing AMD, such as external variables (smoking), internal ones (age, heredity), and numerous stimuli (oxidative stress, inflammation) that subject retinal cells to a detrimental environment. Physiological ocular aging may subsequently progress to pathological deterioration, resulting in dry or wet age-related macular degeneration (AMD). Furthermore, the global aging population may lead to a rise in the prevalence of AMD in the forthcoming years. Table 1 outlines the primary outcomes of clinical trials assessing the effects of fatty acids on AMD.

Current therapies primarily address wet AMD via the intravitreal delivery of anti-VEGF agents. The intravitreal administration of anti-VEGF drugs is extremely effective but considerably invasive, as intravitreal injections can lead to considerable adverse effects. It is not advisable to give these anti-VEGF agents by mouth, because a whole-body blockade of VEGF can lead to poor growth of blood vessels that are essential for the proper functioning of organs. Therefore, in Europe, doctors primarily advise dietary supplements of vitamins, minerals, and omega-3 PUFAs for dry AMD. The disadvantage is that oral administration results in poor absorption of lipophilic compounds [119,120,121], such as fatty acids, which greatly reduces its effectiveness for eye diseases. Various studies have shown that LC-PUFA and VLC-PUFA are significant alternatives for treating AMD, and it would be beneficial to create novel formulations that include these active compounds. Given the reduced preference for intravitreal injections due to their invasive nature and restricted bioavailability through the oral route, it may be advantageous to create an ocular formulation for topical treatment. The tropical treatment has several benefits, including ease of administration, dosage flexibility, alignment with the lifestyle of patients, and improved adherence. The topical approach would consequently mitigate the factors contributing to non-adherence to intravitreal injection therapies.

It is difficult to reach the back part of the eye with topical treatments because of different barriers and the eye’s natural defenses. It is necessary to create an ophthalmic formulation that addresses all these characteristics and facilitates an extended residence duration in the targeted ocular regions. We may produce nanotechnologies, hydrogels, or customized release formulations to achieve this goal. Numerous galenic formulations facilitate the concurrent encapsulation of both lipophilic and hydrophilic substances. A multitude of options for raw materials or combinations is available for wet AMD (a blend of anti-VEGF agents and PUFAs) and dry AMD (PUFAs with vitamins and minerals).

Numerous studies have explored the progress of nanotechnology in eye treatments that are applied directly to the eye to help with problems in the back part of the eye; however, there is not much research specifically focused on AMD using special active ingredients that can reach retinal cells. Additional studies on the treatment of AMD with fatty acids for topical ocular administration may facilitate the development of an effective, permanent therapy for people with AMD.

## 7. Conclusions

Numerous cohort, case–control, and research studies have demonstrated that omega-3 PUFAs, notably DHA and EPA, consistently have a protective effect against AMD, possibly through anti-inflammatory mechanisms and structural roles in the retina. The impacts of SFAs, MUFAs, and omega-6 PUFAs are ambiguous and may differ depending on the fatty acids, dietary context, and genetic influences. Additional meticulously designed intervention trials are necessary to ascertain the ideal dietary fatty acid recommendations for the prevention and management of AMD.

## 8. Methodology of Literature Search

For this review, we performed a MEDLINE/PubMed search for articles published from 1987 to 2025, using the following keywords: “Fatty acids and AMD”, “omega-3 fatty acids and AMD”, “omega-6 fatty acids and AMD”, “saturated fatty acids and AMD”, “monounsaturated fatty acids and AMD”, “long chain polyunsaturated fatty acids & AMD”, “very long chain polyunsaturated fatty acids & AMD”, “AMD” and “desaturases role in long chain poly unsaturated fatty acids”. Current contents and relevant articles on the role of fatty acids in AMD were also obtained using a Google search. Published papers in languages other than English were excluded. We read all 127 articles and did not contact any authors.

## Figures and Tables

**Figure 1 cells-14-00937-f001:**
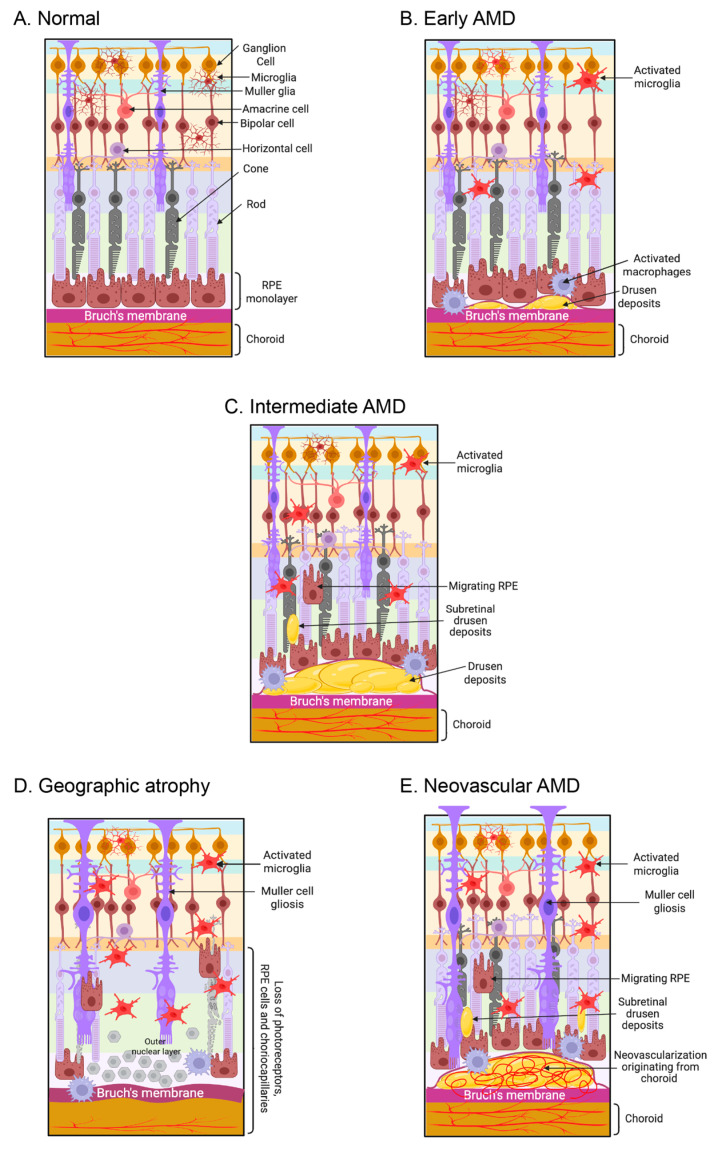
AMD affects retinal photoreceptors, retinal pigment epithelium (RPE), Bruch’s membrane, and choroid choriocapillaris. (**A**) The schematic diagram illustrates a normal retina-RPE-Bruch’s membrane-choroid complex. (**B**) The early phases of AMD are marked by the accumulation of drusen (extracellular debris), activated microglia, and macrophages. (**C**) In the intermediate phase of AMD, accumulation of drusen and subretinal drusen deposits were observed in the retina, along with pathological alterations in the RPE and the migration of RPE cells into the retina. (**D**) Loss of the photoreceptors, RPE, and choriocapillaris are observed in dry AMD. (**E**) In wet AMD, macular neovascularization invades the outer retina, subretinal space, or subretinal pigment epithelium space. The figure is created with www.biorender.com (accessed on 15 March 2025).

**Figure 2 cells-14-00937-f002:**
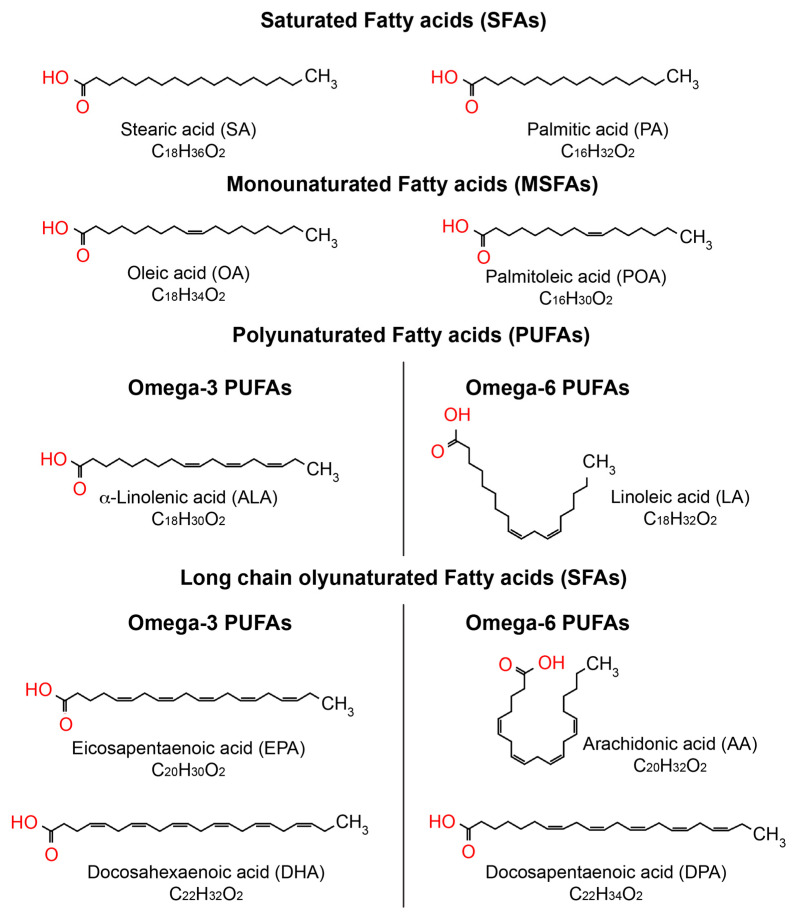
Biochemical representation of the predominant fatty acids in the eye. The chemical structure of fatty acids was created using ChemSpider, https://www.chemspider.com/StructureSearch (accessed on 15 March 2025).

**Figure 3 cells-14-00937-f003:**
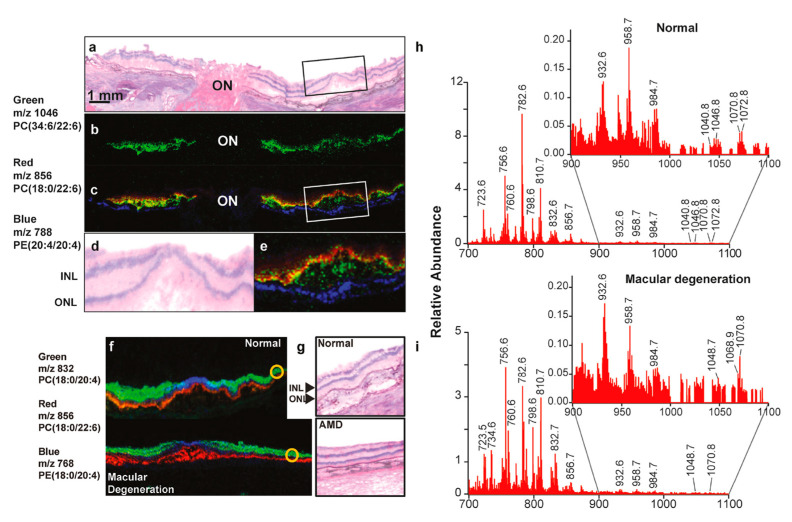
MALDI imaging reveals the distribution of PUFAs within the retinal photoreceptor layer. (**a**) Hematoxylin and eosin staining of the optic nerve (ON) and the fovea of a normal retina. (**b**) MALDI imaging identifies the very long-chain PUFA (m/z 1046, green). (**c**) The composite picture of VLC-PUFA (green) and photoreceptors (red). The colocalization (yellow) of VLC-PUFAs and photoreceptors is seen. The RPE layer is in blue color. (**d**,**e**) Depicts the enlarged boxed regions of (**a**) and (**c**), respectively. INL and ONL denote the inner nuclear layer and outer nuclear layer, respectively. (**f**) MALDI pictures of a normal retina and an AMD retina. The photoreceptor layer is represented in red; the inner retina is shown in green, and the central macular region is depicted in blue. (**g**) The H&E sections show the regions of interest (yellow circles), illustrating the intensity of phospholipids in both normal and AMD retinas. (**h**,**i**) Spectra with mass numbers ranging from 700 to 1100 m/z were acquired from outlying locations, which are indicated by the yellow-circled areas in (**f**). The regions from 900 to 1100 m/z have been expanded as an insert to illustrate variations among the VLC-PUFAs, which are observed between 1000 m/z and 1100 m/z. Reproduced with permission from Gordon et al., Exp Eye Res. 235, 109639 (2023) [103]. Copyright © 2023, Elsevier.

**Table 1 cells-14-00937-t001:** Clinical investigations of fatty acid consumption and AMD.

Study	Year	Dietary Components	Number of Participants	Observations	Reference
US twin study of AMD	2006	Omega-3 fatty acids and fish	681	Reduced risk of intermediate and late AMD	[108]
Polanut study	2007	Total fat, SFA, MUFA, PUFAs, and fatty fish	832	Total, SFA, and MUFA intake induces age-related maculopathy (ARM) risk. PUFA intake has no effect on ARM. Fatty fish intake reduces ARM.	[40]
Blue Mountain Eye Study	2009	Total fat, SFA, MUFA, PUFAs LC-PUFAs, fish, and nuts	2454	A low intake of LA-rich foods and increased intake of fish, nuts, and omega-3 PUFAs offer protection against early AMD.	[38]
Melbourne Collaborative Cohort Study	2009	Total fat, SFA, MUFA, PUFAs, LC-PUFAs, olive oil	6734	Reduced risk of early and late AMD with a diet low in trans-unsaturated fatty acids and high in omega-3 PUFAs and olive oil.	[23]
AREDS	2009	Omega-3 LC-PUFAs	1837	Participants with a high intake of omega-3 LC-PUFAs are 30% less likely to develop central dry and wet AMD.	[14]
Alienor study	2011	Omega-3 LC-PUFAs	1289	Reduced risk of ARM in participants with higher intake of omega-3 LC-PUFAs	[113]
NAT2 study	2013	DHA and EPA	263	Reduced wet AMD with high EPA plus DHA intake	[114]
AREDS 2	2013	Lutien + zeaxanthin, DHA + EPA in addition to Omega-3 LC-PUFAs	4203	The addition of lutein + zeaxanthin, DHA + EPA, or both to omega-3 LC-PUFAs did not reduce the risk of advanced AMD.	[71]
Professional follow-up study + Nurses’ Health Study	2017	DHA and EPA	114850	Increased consumption of EPA and DHA may avert or defer the emergence of intermediate AMD, with little impact on late AMD.	[115]
European Eye Study	2017	Mediterranean diet	5060	Adherence to the Mediterranean diet prevents late-stage AMD.	[116]
Eye-risk consortium	2019	Mediterranean diet	4996	Adherence to the Mediterranean diet reduces the risk of late AMD.	[64]
An ancillary study	2020	Vitamin D3 and omega-3 PUFAs	25871	No significant effect of vitamin D3 or omega-3 PUFAs on AMD incidence or progression.	[117]
Madeos Study	2024	EPA-rich omega-3 PUFAs	21	EPA-rich omega-3 intake may enhance vision in individuals with dry AMD or Stargardt disease.	[118]

## Data Availability

No new data were created or analyzed in this study.

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
