# Peer review of "Implications of Fatty Acids for Age-Related Macular Degeneration: Evidence and Recommendations"

_cells, 2025, doi:10.3390/cells14120937_

Round 1

Reviewer 1 Report

Comments and Suggestions for Authors

General Assessment: This review provides a comprehensive and timely synthesis of current research on the role of fatty acids in the pathogenesis and potential treatment of age-related macular degeneration (AMD). The authors have successfully compiled findings from clinical studies, animal work and cell culture studies, offering valuable insights into how dietary lipids - particularly omega-3 and omega-6 polyunsaturated fatty acids - may influence AMD progression and therapeutic outcomes.

Major comments:

The manuscript includes a wide range of studies, from foundational trials such as AREDS to recent molecular investigations into genes like ELOVL4 and their role in retinal lipid metabolism. However, several citations are misplaced, or relevant literature not cited. Selecting more appropriate references would improve the clarity of this manuscript.

The review is well-structured, with clearly defined sections that address the various types of fatty acids and offer valuable insights into their underlying mechanisms. However, it would be beneficial to include a discussion on the role of Delta-6 Desaturase, in the biosynthesis of long-chain polyunsaturated fatty acids.

The inclusion of supporting figures enhances readability and comprehension, however, Fig.1 could be ameliorated by showing the gliolysis response from the Muller Glia following the loss of the photoreceptors as the AMD progresses. Additionally, the phenotypic changes of microglia associated with the loss of photoreceptors would provide a more comprehensive depiction of the dynamic cellular processes involved in the disease progression.

Minor comments:

Lines 31-32: As of 2025, more than 18 million people in USA are affected with AMD. https://www.visioncenter.org/resources/amd-prevention-statistics/

In summary, this manuscript is a valuable contribution to the vision research field, with a focus on the impact of the diet and implication of fatty acids on AMD. With minor revisions to enhance critical analysis and specificity, it will serve as a useful resource for guiding future research.

Comments on the Quality of English Language

There are typographical errors throughout the text that need to be addressed, such as line 65: PUFAS is missing the closing parenthesis; line 241: “PUFAs” and “in” are fused; line 327: “suc” instead of “such”. 

Several citations in the reference section contain symbols in place of standard alphanumeric characters. A comprehensive review and correction of the references is recommended. Examples: lines 523, 532, 534,568, etc.

Author Response

We express appreciation to the reviewer for their recommendations, and the revised manuscript includes all the reviewer's suggestions, marked in red.

Major comments:

  1. The manuscript includes a wide range of studies, from foundational trials such as AREDS to recent molecular investigations into genes like ELOVL4 and their role in retinal lipid metabolism. However, several citations are misplaced, or relevant literature not cited. Selecting more appropriate references would improve the clarity of this manuscript.

Answer: We thank the reviewer for his/her suggestions, and we have incorporated more references to the revised manuscript (Please refer to references 1, 5, 35, 39, 55, 69, 71, 72, 73, 76, 83, 84, 115, 116, 117, and 118).

  1. The review is well-structured, with clearly defined sections that address the various types of fatty acids and offer valuable insights into their underlying mechanisms. However, it would be beneficial to include a discussion on the role of Delta-6 Desaturase, in the biosynthesis of long-chain polyunsaturated fatty acids.

Answer: We thank the reviewer for his query, and now a discussion on the role of Delta-6 Desaturase is included in the revised manuscript (please refer to page 6, lines 193-201).

  1. The inclusion of supporting figures enhances readability and comprehension; however, Fig.1 could be ameliorated by showing the gliolysis response from the Muller Glia following the loss of the photoreceptors as the AMD progresses. Additionally, the phenotypic changes of microglia associated with the loss of photoreceptors would provide a more comprehensive depiction of the dynamic cellular processes involved in the disease progression.

Answer: We thank the reviewer for these important suggestions, and we have now revised the Figure 1 to included reviewer #1 suggestions (Please refer to Figure 1).

Minor comments:

1.Lines 31-32: As of 2025, more than 18 million people in USA are affected with AMD. https://www.visioncenter.org/resources/amd-prevention-statistics/

Answer: We have now included the updated information in the revised manuscript (please refer to page 1, line 34)

  1. There are typographical errors throughout the text that need to be addressed, such as line 65: PUFAS is missing the closing parenthesis; line 241: “PUFAs” and “in” are fused; line 327: “suc” instead of “such”. 

Answer: We have corrected it in the revised manuscript.

  1. Several citations in the reference section contain symbols in place of standard alphanumeric characters. A comprehensive review and correction of the references is recommended. Examples: lines 523, 532, 534,568, etc.

Answer: We have corrected it in the revised manuscript.

Reviewer 2 Report

Comments and Suggestions for Authors

See attach

Author Response

We express appreciation to the reviewer for their recommendations, and the revised manuscript includes all the reviewer's suggestions, marked in red.

Major comments:

1. The manuscript contains redundant information, especially in sections 3-5 where epidemiological findings are often repeated and discussed several times in similar terms.

Answer: We thank the reviewer for his/her suggestions, and we have now removed the repeated and redundant information from the revised manuscript.

2. Although this 1s a review, the methodology of inclusion/exclusion of literature 1s not discussed. It is suggested to include a brief section explaining the search strategy: databases, keywords, inclusion/exclusion criteria, and time frame covered.

Answer: We thank the reviewer for his/her query and now included “Methodology of Literature Search” section in the revised manuscript (please refer to page 12, lines 398-407).

3. Some phrases are informal or lack scientific precision, e.g., "people do not like the intravitreal route," "blood flow problems in the choroid," or "diets rich in meat and dairy products contain a lot of omega-6." It is suggested that these expressions be replaced with formal scientific expressions (e.g., "intravitreal injections are less favored because of invasiveness" or "diets rich in omega-6 PUFAs have been associated with elevated inflammatory responses").

Answer: We thank the reviewer for his/her suggestions, and we have now included it in the revised manuscript (please refer to page 11, lines 369-370, page 5, lines 139-141, page 7, lines 250-251).

4. Another issue is the section on very long-chain polyunsaturated fatty acids, which is rich in descriptive detail but lacks integration with the broader pathology of AMD and therapeutic goals. Clarifying how VLC-PUF A deficits contribute mechanistically to photoreceptor degeneration and whether dietary strategies can realistically compensate for ELOVL4 or ELOVL2 dysfunction.

Answer: We thank the reviewer for his/her thoughtful suggestions, and we have now included it in the revised manuscript (please refer to page 9, lines 307-319).

5. Some statements are made without clear citations (e.g., fish intake reducing AMD risk is repeatedly stated but not always supported by the same references). Ensure that any claims, especially epidemiological claims, are referenced to the literature.

Answer: We have now corrected it in the revised manuscript.

Minor revisions:

1. Several sentences are grammatically incorrect. Examples: "Fatty acids are essential for the human body ... " - "Fatty acids are essential for the human body ... "-"AMD is characterized by degeneration of the macula and its photoreceptors, which are composed of numerous fatty acids." - Reword to clarify that photoreceptor membranes are lipid-rich, not "composed of fatty acids."

Answer: We have corrected it in the revised manuscript (please refer to page 3, line 90, page 6, lines 205-206).

2. Use consistent naming for disease subtypes: or "wet AMD" vs. "neovascular AMD" - choose one and use it consistently. or "dry AMD" vs. "geographic atrophy" - clarify whether these are synonyms and use a standardized designation throughout the text.

Answer: We have corrected it in the revised manuscript

3. Figure 1: Improve clarity of legends; avoid ambiguous expressions such as "B and C, the early and middle stages."

Answer: We have rewritten it in the revised manuscript (please refer to page 3, lines 58 and 59).

4. Table 1: consider reformatting to ensure readability. Abbreviations such as "ARM" should be clearly defined (assumed to mean "Age-related maculopathy"?).

Answer: We have corrected it in the revised manuscript (please refer to Table 1).

5. Many minor typos are present: "poly unsaturated fatty acids" - "polyunsaturated fatty acids""AMD is also called as geographic atrophy" - remove "as" Some abbreviations (e.g., RPD, ONL, INL) are used without definition when they first appear. Ensure that all abbreviations are defined when they are introduced.

Answer: We have corrected it in the revised manuscript.

Reviewer 3 Report

Comments and Suggestions for Authors

The paper's title and aims are auspicious. Unfortunately, the review lacks novelty. It merely summarizes the existing literature without offering new insights or a unique perspective that could guide future development. Additionally, it fails to provide a more in-depth discussion and conclusion directly connected to the manuscript's goal, which would help highlight the manuscript's contribution to advancements in the field.

Therefore, I do not recommend its publication in its current form.

Other comments that could enhance the presentation of the manuscript include the following:

  1. To enhance the impact of a review paper, it is essential to go beyond merely summarizing existing literature. I suggest adding new insights, such as presenting a unique perspective and offering a clear direction for future research.
  2. Include a concluding sentence in the abstract.
  3. In Figure 1, I suggest that instead of including the names of the cellular structures in all the images, including them in the first image and the changes that define the states in the others would be better.
  4. The conclusion section functions more like a discussion section. It is too long for a conclusion but too short for a discussion. Please revise and modify it to align with the title and goals of the paper.
  5. It is essential to note that, although many references are cited, only 17% are from the last five years. This suggests that the topic is not relevant today.
Comments on the Quality of English Language

 The English language is adequate. However, it could be improved to express the research more clearly.

Author Response

We express appreciation to the reviewer for their recommendations, and the revised manuscript includes all the reviewer's suggestions, marked in red.

1. To enhance the impact of a review paper, it is essential to go beyond merely summarizing existing literature. I suggest adding new insights, such as presenting a unique perspective and offering a clear direction for future research.

Answer: As per the reviewer’s suggestions, we have summarized it in the revised manuscript (please refer to page 5, lines 155-159, page 6, lines 174-182, page 7, lines 240-248, page 9, lines 320-345).

 2. Include a concluding sentence in the abstract.

Answer: We have included it in the revised manuscript (please refer to page 1, lines 22-25).

3. In Figure 1, I suggest that instead of including the names of the cellular structures in all the images, including them in the first image and the changes that define the states in the others would be better.

Answer: We have corrected it in the revised manuscript (please refer to Figure 1).

4. The conclusion section functions more like a discussion section. It is too long for a conclusion but too short for a discussion. Please revise and modify it to align with the title and goals of the paper.

Answer: We have included it in the revised manuscript (please refer to page 12, lines 389-396).

5. It is essential to note that, although many references are cited, only 17% are from the last five years. This suggests that the topic is not relevant today.

Answer: We have included new references in the revised manuscript (please refer to references 1, 5, 35, 39, 55, 69, 71, 72, 73, 76, 83, 84, 115, 116, 117, and 118).

Reviewer 4 Report

Comments and Suggestions for Authors

What are the specific underlying mechanisms of fatty acid metabolism differences between dry and wet AMD?
Are the actions of different fatty acids on AMD progression dose-dependent?
How do genetic variations (e.g., ELOVL4 or CFH gene polymorphisms) influence responses to dietary fatty acid in AMD?

What is the function of the gut microbiome in modulating fatty acid metabolism and how does it impact AMD?
Are there unpublished or ongoing clinical trials that compare topically or newer delivery forms of PUFAs for AMD?
Do comorbidities like diabetes or cardiovascular disease affect the efficacy of fatty acids in preventing or progressing AMD?

Are there comparative bioavailabilities of different PUFA formulations (e.g., fish oil, algal oil, esterified) in retinal tissue?
Is there a critical age window for dietary fatty acid supplementation to postpone the onset or slow the development of AMD?
What are the side effects or risks of long-term administration of high doses of fatty acids in older people?
How does whether dietary fatty acids are from animal or plant sources influence their prevention of AMD?

Author Response

We express appreciation to the reviewer for their recommendations, and the revised manuscript includes all the reviewer's suggestions, marked in red.

1. What are the specific underlying mechanisms of fatty acid metabolism differences between dry and wet AMD? Are the actions of different fatty acids on AMD progression dose-dependent?
How do genetic variations (e.g., ELOVL4 or CFH gene polymorphisms) influence responses to dietary fatty acid in AMD?

Answer: We thank the reviewer for his/her suggestions, and we have now included it in the revised manuscript (Please refer to page 7, lines 229-232, page 9, 307-319).

2. What is the function of the gut microbiome in modulating fatty acid metabolism and how does it impact AMD? Are there unpublished or ongoing clinical trials that compare topically or newer delivery forms of PUFAs for AMD? Do comorbidities like diabetes or cardiovascular disease affect the efficacy of fatty acids in preventing or progressing AMD?

Answer: We thank the reviewer for his/her thoughtful suggestions, but these are beyond the scope of this article.

3. Are there comparative bioavailabilities of different PUFA formulations (e.g., fish oil, algal oil, esterified) in retinal tissue? Is there a critical age window for dietary fatty acid supplementation to postpone the onset or slow the development of AMD? What are the side effects or risks of long-term administration of high doses of fatty acids in older people? How does whether dietary fatty acids are from animal or plant sources influence their prevention of AMD?

Answer: As per the reviewer’s suggestions, we have looked for the literature and were unable to find more conclusive findings on the role of fatty acids on these topics.

Round 2

Reviewer 2 Report

Comments and Suggestions for Authors

No further comments.

Reviewer 4 Report

Comments and Suggestions for Authors

The paper can be accepted in its present form.